# Cross-Latitude Behavioural Axis in an Adult Damselfly *Calopteryx splendens* (Harris, 1780)

**DOI:** 10.3390/insects13040342

**Published:** 2022-03-31

**Authors:** Maria J. Golab, Szymon Sniegula, Tomas Brodin

**Affiliations:** 1Institute of Nature Conservation, Polish Academy of Sciences, 31-120 Krakow, Poland; 2Department of Wildlife, Fish and Environmental Studies, Swedish University of Agricultural Sciences, 90187 Umeå, Sweden; tomas.brodin@slu.se

**Keywords:** Odonata, latitudinal gradient, boldness, activity, courtship, behavioural syndrome

## Abstract

**Simple Summary:**

Animals adapt to the environment they live in. If the environment changes, animals usually adapt behaviourally as a first response. By studying behavioural profiles across long distances, we can detect environmental change reflected in shifts in behavioural profiles. This study examined variation in three behavioural axes: activity, courtship and boldness, and the association between these behaviours, i.e., behavioural syndromes, across three damselfly populations along a latitudinal gradient (i.e., climatic gradient). Our study organism was the temperate damselfly *Calopteryx splendens*. We predicted that behavioural expressions would gradually increase from southern to northern regions. This is because northern animals should compensate behaviourally for a brief and cold breeding season (i.e., time constraint). Activity was the only behaviour feature positively associated with latitudinal gradient. Courtship effort was highest in the central region, whereas boldness values were highest in the north but did not differ between central and south. In the southern region, an activity–boldness and a courtship—boldness syndrome were present, and in the northern region, only an activity–boldness syndrome was found. Our results confirm that environmental variability in biotic and abiotic factors across studied latitudes generates regional differences in behavioural profiles, which do not always follow latitudinal gradient.

**Abstract:**

Behavioural variation is important for evolutionary and ecological processes, but can also be useful when predicting consequences of climate change and effects on species ranges. Latitudinal differences in behaviour have received relatively limited research interest when compared to morphological, life history and physiological traits. This study examined differences in expression of three behavioural axes: activity, courtship and boldness, and their correlations, along a European latitudinal gradient spanning ca. 1500 km. The study organism was the temperate damselfly *Calopteryx splendens* (Harris). We predicted that the expression of both behavioural traits and behavioural syndromes would be positively correlated to latitude, with the lowest values in the southern populations, followed by central and the highest in the north, because animals usually compensate behaviourally for increasing time constraints and declining environmental conditions. We found that behavioural expression varied along the latitudinal cline, although not always in the predicted direction. Activity was the only behaviour that followed our prediction and gradually increased northward. Whereas no south-to-north gradient was seen in any of the behavioural syndromes. The results, particularly for activity, suggest that climatic differences across latitudes change behavioural profiles. However, for other traits such as courtship and boldness, local factors might invoke stronger selection pressures, disrupting the predicted latitudinal pattern.

## 1. Introduction

Behavioural adaptations are often an individual’s first response to an environmental change [1]. Behaviours can also evolve faster and more dynamically than life history, physiology and morphology [2,3]. Hence, behavioural traits can be excellent indicators of environmental disturbance, i.e., an early warning system [4,5]. It has been shown that some species can buffer unfavourable environmental changes by adjusting behaviour, whereas other species cannot [6,7]. There are several reasons for why the latter may arise. The trait variance can be too low [8], evolution of a trait may be constrained by genetic correlations among traits (behavioural syndromes) [9] or adaptive behaviours became maladaptive when organisms faced anthropogenic disturbances, such as, for example, some types of artificial surfaces attracting water-seeking insects [10]. Additionally, behavioural responses to climate change may trigger changes in phenology [11] or physiology [12].

Cross-population studies of behavioural traits have gained extensive research interest. Studies on one species provide stronger ecological and evolutionary insights of behavioural variability than cross-species comparisons [13]. Especially interesting are behavioural comparisons along different environmental gradients. For example, studying organismal behaviour along latitudinal gradients can help us understand how organisms are adapted to variable climates and predict how organisms will respond to rapid climate change [14]. Despite this, comparably few studies have reported behavioural differences along latitudinal gradients [15,16].

Changes in behavioural expressions along environmental gradients may be gradual or can display a threshold. For instance, studies on stream invertebrates showed that progressive drought caused faster and gradual responses of locomotion or dispersal traits, while morphological and physiological traits responded only after the water dried up completely (i.e., threshold) [17]. It is therefore important to identify critical thresholds when they are present, in order to draw reliable conclusions on future trait changes.

Boldness and activity are traits intensively studied by behavioural ecologists [18,19]. These traits play an important role in species range shifts caused by climate change [20,21]. For example, it has been shown that nest foundresses of invasive *Vespa velutina* were bolder and more active compared to queens from native ranges [22]. In addition, these traits are often correlated, creating so-called behavioural syndromes, which is a correlation of two or more behavioural traits responding synchronously to a variety of contexts [23,24]. However, the absence of behavioural syndromes (e.g., activity–boldness) has also been reported [25], and traits were decoupled by different strengths of natural and sexual selections in different environments [9,25].

Traits that are subject to sexual selection are also sensitive to environmental conditions [26]. For example, male newts *Mesoriton alpestris* rely on tactile and chemical signals to attract females when light conditions are poor. However, visual display enhance males’ mating success in a better lit environment, indicating that sexual selection of male newts is environmentally dependent [27]. If natural and sexual selection on the same trait (or syndrome) are in opposition, it is important to know which of the two selections is more important for shaping the behavioural profile [28]. However, this is not always easy to quantify.

Here, we studied latitudinal differences in activity, courtship, boldness (i.e., behavioural axes) and their correlations (i.e., behavioural syndromes) in natural populations of the temperate damselfly, *Calopteryx splendens*. This species is an excellent study organism because it has a wide latitudinal distribution and displays conspicuous behaviours [29]. We hypothesise that expression (i.e., variation in the trait values) of behavioural axes and behavioural correlations will be positively correlated with latitude, with progressively increased activity, courtship effort and boldness northwards. This is because time constraints for growth and breeding are more severe northwards, i.e., the climate becomes colder and the breeding season shorter [30]. Hence, natural and sexual selections should favour behavioural expressions and syndromes that compensate for these environmental constraints [4].

## 2. Materials and Methods

### 2.1. Study Species

*Calopteryx splendens* (Figure 1a) is a common Eurasian damselfly, inhabiting lowland rivers and slow-flowing streams. Its European latitudinal range extends from northern Scandinavia to the coast of the Mediterranean Sea. Hence, the species experiences a major latitudinal gradient in ambient temperatures and breeding season length [31]. The species exhibits strong site fidelity [32] and displays a wide range of sexual and territorial behaviours. Males adopt either territorial or non-territorial strategies, and territorial males fight with conspecifics to gain access to a territory (i.e., patch of floating vegetation). Once a male obtains a territory, he starts to defend it by patrolling along its ranges and chasing away any intruders approaching the area [33]. Patrol flight of a territorial male is a rapid circular flight along the borders of his territory [34,35]. Non-territorials either patrol over the larger sections of a river, with no particular attachment to one place, or can exhibit satellite tactics, i.e., when a male focuses on a selected area but does not own a territory and hence does not defend it [33]. Once a female approaches a territory, the territorial male starts to court her by showing his territory quality (i.e., patrol flights and so-called dive display, that is, alighting on the water surface for a few seconds) and changing wing-beating frequency [36,37,38].

### 2.2. Experimental Setup

Behavioural data were collected in the summer of 2016 on small lowland rivers at three regions: Hungary, Poland and Sweden (hereafter called southern or low latitude, central and northern or high latitude region, respectively), covering a distance of ca. 1500 km (Figure 1b). In each region, two allopatric (one calopterygid species only) *C. splendens* populations (in total 6 populations and 6 river sections, Figure 1b, Table 1) were studied during the peak of the mating season (Table 1). Field studies were conducted on 50 m river sections, between 10:00 and 15:00 hours, under favourable and comparable weather conditions (sunny, low-wind, warm days: 24–29 degrees Celsius [37,39,40]). The population densities were comparable among the sites and assessed based on the numbers of captured animals on the studied river section (Table 1). Prior to the experiments, all males present at the studied river section were caught and individually marked (3-digit number) with a white marking pen (Figure 1a). After the marking session (ca. 1 h), damselflies were left undisturbed for at least 30 min until all males resumed natural activity. Next, several territorial individuals were chosen for further experiments on each study day based on the following criteria: a male defended a single territory [34], the territory is at least 3 m away from others already selected for experiments, the area is fully exposed to sunshine so that the microclimatic conditions were similar. We ran three behavioural assays: (1) activity, (2) courtship, and (3) boldness (Table 2). Activity was monitored in familiar low-risk conditions [41] during a 20 min observation period, and two traits were measured: patrol flights and male chases. For the courtship assessment, we counted the number of courtship displays by a territorial male during a 20 min observation period. Boldness was measured during a simulated predator attack, which is a common method for boldness estimations [42,43,44]. Based on our previous experience, we designed a standard method for the simulated predator attack. An artificial bird attached to a fishing rod (4.2 m long) was moved at a speed of approximately 1 m/s downstream toward the territorial male resting on his perching site. The rod was handled by a person standing in the middle of the river upstream of the territorial male, and the attack simulation lasted ca. 3–5 s. Once the simulated predator attack was carried out, the bird decoy was retracted back (when the decoy was ca. 5 cm from the male perching place) to the observer and kept motionless until the damselfly returned to its territory. All damselfly males reacted to the attack when the approaching bird decoy was closer than 0.5 m. Three traits were measured during the attack: immobility, first move and time to return (Table 2). The behavioural assays were then repeated 10 minutes after the end of the boldness trial. The mean trait values for each behaviour were later used in the statistical analyses. Note that previous studies on *C. splendens* showed repeatability in boldness [40] (Table 2). After the last behavioural trial ended, males were caught, and their size (i.e., head width and thorax length) was measured with a digital calliper to the nearest 0.1 mm.

### 2.3. Statistical Analyses

The statistical analyses were performed using IBM SPSS Statistics for Windows, version 26 [45]. To reduce number of variables a principal component analysis (PCA) was applied for the traits on the boldness axis. Both activity and body size were calculated by averaging the Z-scores for the two variables within each trait. Courtship was measured as a single variable (Table 2).

Generalized linear mixed models (GLMM) with Poisson (courtship), Gamma (boldness) and normal (activity) distributions were used to test for differences in three behavioural axes: PC1 for boldness, averaged Z-scores of activity, courtship and body size between the three regions. Population within region was used as a random effect. Pairwise contrast comparisons were used to test for differences between the regions. In order to assess the presence of behavioural syndromes, Spearman’s correlations between PC1 value of boldness and averaged Z-scores of activity and courtship were used. Please note that boldness values were measured as time of reaction to our treatment, the bolder the individual was, the shorter the measured time. Therefore, in order to facilitate the interpretation of correlations, the boldness values have been inverted. In all models, we included body size as a covariate. Statistical significance was established at *p* < 0.05.

## 3. Results

### 3.1. Descriptive Statistics

In total, we studied 140 males: 31 males in Sorb population, 23 in Ikva, Kard and Arby populations, 21 in Ruda and 19 males in Usze (Table 1). Activity was measured for 135 males, courtship for 135 males, boldness for 125 males and body size for 123 males.

### 3.2. Behavioural Syndromes

There was a significant positive correlation between activity and boldness in southern (r = 0.35, *p* = 0.023) and northern regions (r = 0.35, *p* = 0.024), but no correlation in the central region (r = 0.21, *p* = 0.207). Courtship was positively correlated with boldness in the south only (r = 0.34, *p* = 0.024). No other behavioural correlations were significant.

### 3.3. Body Size

GLMM models showed that body size was positively correlated with latitude, (F = 9.81, *p* < 0.001, Table 3, Figure 2d). Since body size had no significant effect on activity (F = 0.65; *p* = 0.587) or boldness (F = 0.79, *p* = 0.505), it was not included in the final models for those behaviours. In contrast, body size did affect courtship (F = 4.28, *p* = 0.041) so that the largest males in the northern region performed fewest courtships (Figure 2b,d). There was no significant interaction between body size and region for courtship (F = 2.56, *p* = 0.059).

### 3.4. Behaviours

The first principal component (PC1) explained 83.06% of the variation in boldness (eigenvalue = 2.49) and had the highest loadings on immobility (0.96), first move (0.97) and time to return (0.79). The other PCs all had eigenvalues below 0.5 and were hence not used in any analysis.

Activity was the only behavioural trait that varied predictably with latitude; damselflies expressed the lowest activity in the south and the highest in the north (F = 25.02, *p* < 0.001, Table 3, Figure 2a). Courtship, on the other hand, was lowest in the north (F = 11.17, *p* < 0.001, Table 3, Figure 2b), followed by central and low latitudes that did not differ in trait expression. Boldness was significantly higher in the north compared to the south and central populations (F = 4.87, *p* = 0.009, Table 3, Figure 2c). Boldness of damselflies from the southern and central regions did not differ (Table 3, Figure 2c).

## 4. Discussion

We examined expressions of three important behavioural traits along a latitudinal gradient and found that only activity changed as predicted and increased gradually northwards, whereas courtship (highest values in central region) and boldness (highest in the north) showed changes potentially triggered by thresholds. We found a positive activity–boldness behavioural syndrome in low and high latitudes only. Courtship was positively correlated with boldness in the southern populations, but no correlation between the traits was found in the central or northern populations. These results partially support our hypothesis and suggest that the short and cold breeding season at high latitudes selects for more active and bold individuals, whereas courtship is probably shaped by social interactions in the central populations.

Latitudinal differences in activity have been studied in other species, for example in the common spreadwing (*Lestes sponsa*, Hansemann), and similarly to our results, showed a clear positive latitudinal gradient [46]. In contrast, a study on eastern mosquitofish (*Gambusia holbrooki*, Girard) showed no geographical variation in activity [15]. We attribute the latter pattern to the fact that, contrary to temperate populations of *C. splendens* and *L. sponsa*, subtropical populations of mosquitofish are not exposed to ecologically meaningful latitudinal differences in breeding time constraints. We suggest that a gradual increase in activity (here connected to territory defence) towards the north might be a behavioural compensation for a brief breeding season at high latitudes [47]. By increasing their activity in territorial defence, damselfly males prevent their conspecifics from territory takeover (social interaction) [33,34], and as a result, receive more copulations and hence, higher fitness [48,49].

Boldness did not change gradually with latitude. Instead, we suggest that boldness changed at a threshold, that is, an environmental critical value (Figure 2c). High latitude populations, situated close to the northern range margin [29,50], differed from the other regions and showed the highest boldness values. Since *C. splendens* is currently expanding northwards due to global warming [51], marginal populations are probably predisposed for dispersal. Previous results have indicated that risk-taking individuals are more prone for dispersal [29,50], as was also recently shown in dark-eyed junco (*Junco hyemalis*, L.) [52]. Based on these studies, our results suggest that high-latitude populations of *C. splendens* have high capacity to expand north because of their bold behaviour.

In many insect species, including *C. splendens*, courtship plays a crucial role in male mating success [36], and we hypothesised that high-latitude populations should express the highest values in male courtship in order to compensate for a brief breeding season. Interestingly, our results did not support the hypothesis, showing a non-linear relationship between latitude and courtship, with the highest trait values in the central region and lower expressions in northern populations. One of the explanations could be higher receptiveness of northern females, as has been shown in another damselfly species, *L. sponsa* [53]. Under such conditions (more receptive females), investment into courtship would be a waste of energy for males. If this is present also in *C. splendens*, high-latitude females probably choose their mates predominantly based on their boldness and body size, whereas central and low latitude females asses male courtship effort [54,55].

In general, the stronger the behavioural syndrome (higher correlation), the tighter the relationship between two different behavioural traits [24]. Here, behavioural syndromes were only moderately strong (*r* between 0.34–0.35), indicating that the behavioural traits could, at least to some extent, change their expressions rather independently, depending on the environment experienced. Nonetheless, it has been suggested that even weak trait associations can have important ecological and evolutionary implications [24]. Below we discuss possible causes and consequences of the behavioural correlations found here.

The positive activity–boldness syndrome is commonly found in animal populations [16,56,57], and were here only present in high- and low-latitudes populations. The lack of behavioural syndrome in the central populations could be a result of low boldness in the central region combined with high levels of social interactions in *C. splendens*, inducing higher activity (discussed above). The low boldness could be explained by the fact that the relatively high activity (during territory defence and courtship) leads to increased exposure to predators, so that shy males (low boldness) are favoured by natural selection in central populations. An alternative explanation is that a lower predation pressure at the *C. splendens* mating sites in central region leads to the disruption of the behavioural syndrome. Such a disruption was found in three-spined stickleback (*Gasterosteus aculeatus*), which showed aggressive–active behavioural syndrome in ponds were predators were present and showed no such association in waterbodies with no predators [16].

The positive courtship–boldness syndrome, which we noted only in southern populations, might be explained by a relatively strong sexual selection on both courtship and boldness. Females may choose mates based on their personality, as has been shown for example in zebra finches [58]. Therefore, boldness might be a sexually selected trait in southern populations of *C. splendens*, despite the potentially strong bird predation pressure in the south [59]. On one hand, courting males exposed themselves to predators [36], but on the other hand, the bolder the male is, the more likely he is to display courtships and increase his fitness. Additionally, it has been shown that under some circumstances, bold individuals may experience lower predation, if bold and fast reactions help to avoid predators [60]. Alternatively, predation pressure might be moderate at the southern populations (which were located far north of the southern species range margin where the predator diversity would be potentially the highest [59], Figure 1), and hence not affect observed boldness expressions.

Body size differences along the latitudinal gradient can be explained by the variable number of generations per year (voltinism) in the study species. In the northern region, the species completes one generation within two or three years (semi- or parti-voltine), whereas in the central and southern regions, the species completes one or two generations within a year (uni- or bi-voltine [61]). Voltinism depends on environmental conditions such as temperature, and hence the length of the growth season, and genetic components that shape seasonal regulation of larval development and growth [62,63]. At the latitude where the voltinism changes, for example, from univoltine to semivoltine, individuals may increase their body size at emergence because of an extra season available for larval development and growth [64,65]. In our study, body size gradually increased northwards. We suggest that the size increase is caused by changes in voltinism. However, other factors, e.g., antagonistic interactions, might also affect insect body size across latitudes [66], but this needs further study.

Although a larger adult body size demands longer time for heating and movement activation [67], it may also keep temperature [68] and metabolic rate stable for longer, and extend the maximum time spent active [69,70]. These body-size-related behavioural characteristics seem to be adaptive for damselflies that experience seasonal time and temperature constraints for breeding, such as the damselflies in the northern populations studied here.

To our knowledge, this is the first study investigating multiple behavioural traits and behavioural syndromes in natural populations of a temperate insect along a latitudinal gradient. We found a clear latitudinal gradient in activity of adult *C. splendens*. The other results suggest that regional and local differences in environmental stressors and biotic factors influencing sexual selection (also social interactions) may determine thresholds for behavioural change influencing the behavioural patterns of *C. splendens* [16]. Additional studies examining whether the observed behavioural patterns are evolved or plastic will be of great importance for predicting future shifts in behavioural profiles, and as such, the resilience of temperate insects in the face of climate change.

## Figures and Tables

**Figure 1 insects-13-00342-f001:**
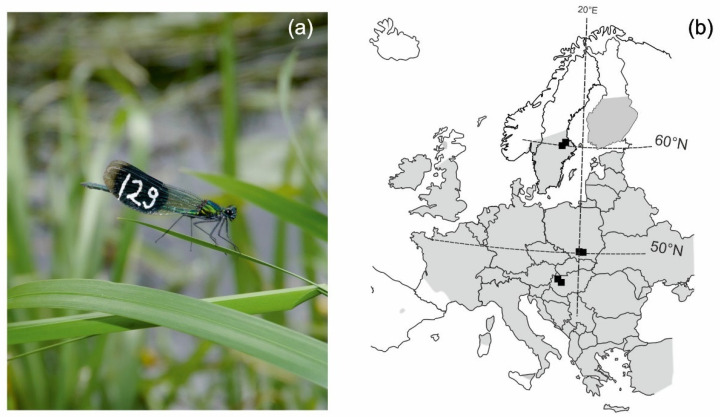
(**a**) Marked territorial male *Calopteryx splendens* and (**b**) location of the sampling sites (black squares) across the distribution of the species (grey shade) in Europe (modified from [29]).

**Figure 2 insects-13-00342-f002:**
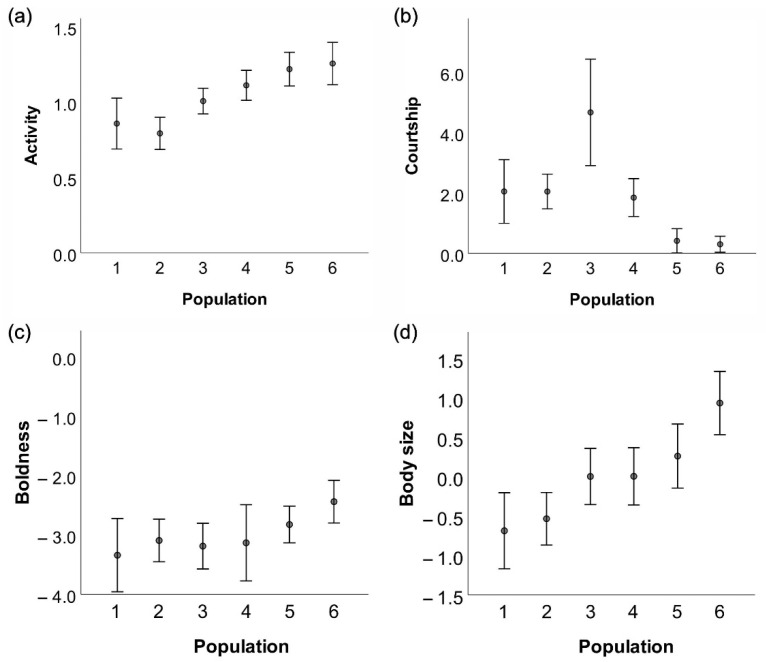
Means and 95% confidence intervals of activity (**a**), courtship (**b**), boldness (**c**), and body size (**d**) of *Calopteryx splendens* between six populations: 1-2 (southern region), 3-4 (central region), and 5-6 (northern region).

**Table 1 insects-13-00342-t001:** Sampling sites (six populations) and sample sizes of *Calopteryx splendens* males in three regions: southern, central and northern.

Region	Population	Coordinates	Sampling Dates	Sample Size
Southern	1	47.59087° N 16.75141° E	27–29 July	23 males
	2	47.573524° N 17.000544° E	31 July–3 August	23 males
Central	3	50.0769° N 19.85464° E	21–23 July	21 males
	4	50.0805° N 20.62278° E	8–11 July	19 males
Northern	5	59.71278° N 16.65955° E	29–30 June	23 males
	6	59.91231° N 16.87902° E	26–28 June	32 males

**Table 2 insects-13-00342-t002:** Description of behavioural traits measured in three behavioural axes of *Calopteryx splendens* from southern, central and northern region.

Behavioural Axis	Trait Measured	Description
Activity	Patrol flights	Number of patrol flights performed during observation period
	Male chases	Number of chases of intruder males approaching or passing a territory defended by a focal territorial male
Courtships	Courtship	Number of courtship displays performed during 20 min observation period
Boldness	Immobility	Total time [s] of the territorial male remaining motionless after the predatory attack simulation. The motionless periods between consequtive movements toward the territory are summed up
	First move	Time [s] until the territorial male first moved after the motionless behaviour
	Time to return	Time [s] passed until the territorial male returned to his territory after the predatory attack. The maximum time the observer waited for the male to come back was 180 s

**Table 3 insects-13-00342-t003:** Post hoc pairwise comparisons of Calopteryx splendens male behaviours and body size between three regions: southern (S), central (C) and northern (N).

Trait	Pairwise Contrast	Estimate	t	df	*p*
Activity	S–C	−0.18	−3.59	131	<0.001
	S–N	−0.34	−7.07	131	<0.001
	C–N	−0.16	−3.19	131	0.002
Courtship	S–C	−0.96	−0.89	131	0.372
	S–N	1.69	2.74	131	0.007
	C–N	0.96	2.98	131	0.003
Boldness	S–C	0.05	0.22	121	0.823
	S–N	0.57	2.81	121	0.006
	C–N	0.52	2.49	121	0.014
Body size	S–C	−0.54	−2.12	120	0.031
	S–N	−1.07	−4.43	120	<0.001
	C–N	−0.54	−2.17	120	0.032

## Data Availability

Data are available in RepOD at https://doi.org/10.18150/4WHGD8 (accessed on 3 February 2022).

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
