# Peer review of "Cross-Latitude Behavioural Axis in an Adult Damselfly Calopteryx splendens (Harris, 1780)"

_insects, 2022, doi:10.3390/insects13040342_

Round 1

Reviewer 1 Report

The study of Golab et al. entitled Cross-latitude behavioural axis in an adult damselfly Calopteryx splendens investigated variation and mutual relationship of three behavioural traits (activity, courtship and boldness) of the damselfly species, C. splendens, along a latitudinal gradient (between Sweden, Poland and Hungary). The authors recorded a clear latitudinal gradient in species activity, while two other behaviour traits, courtship and boldness, might be more significantly influences by the local factors. Studies investigating multiple behavioural traits and behavioural syndromes in natural populations of temperate insects along a latitudinal gradient are rare and those results are interesting and novel, and I would love to see them published. Nevertheless, there are several points that need to be addressed, and one of them is the English language revision by the native speaker. In some cases, I was not completely sure what the authors wanted to say, I corrected some parts/suggested some modifications, but there are still many left to be revised. Additionally, some comments and suggestions are presented below:

Title – please put the species name in italic, and add authority

Line 16 – from southern to northern region

Line 17 - This because northern animals – This IS because...

Line 18 - Activity was the only behaviour positively associated.... – I would rephrase this as Activity was the only behaviour FEATURE positively associated...

Lines 19-20 - Courtship was highest in the central region whereas boldness was highest in the north but did not differ between central and south. – what does this mean? What does the highest courtship, of the highest boldness mean?

Lines 20-21 - An activity–boldness syndrome was only present in the southern and northern region, and courtship was associated with boldness in low latitudes only. – it would be good to refer to your regions in one way within the sentence, i.e. if you speak about the southern populations when explaining one behavioural trait, then do not mention low latitude when you are talking about some other investigated trait.

Lines 23 - which not always follow latitudinal gradient. – you mean which do not always follow latitudinal gradient?

Line 26 - Latitudinal differences in behaviour has received relatively limited interest – Latitudinal differences in behaviour HAVE received—also, you mean research interest?

Line 30 – I would suggest using the authority when you mention the species for the first time, in abstract and in the text

Keywords: I would suggest replacing the species name, Calopteryx splendens, with Odonata, as you already have species name in the title

Line 50 - Cross-population studies of behavioural traits have gained large interest – better say have gained an extensive research interest….

Line 50 - These within species studies – what does this mean, please rephrase

Line 52 - Behavioural comparisons of specific interest are those along different environmental gradients – you mean Specifically interesting are behavioural comparisons along different environmental gradients?

Line 66 - invasive signal crayfish – please add latin species name in the brackets

However, absence of behavioural syndromes (e.g. activity – boldness) have also been reported [19]. In that study the traits were decoupled by different strengths of natural and sexual selections in different environments [20]. – when you say „in that study“ you refer to the reference no. 19 or 20?  

Lines 64 – 80 – are there maybe similar examples in aquatic insects?

Line 84 - and conspicuous behavioural displays… - you mean and displays conspicuous behaviour?

Lines 93-94 – please add reference

Line 95 - the species experience à the species experiences, or goes through

Line 96 - The species exhibit strong site fidelity – you are talking about one species, so it has to be the species EXHIBITS

Line 100 – a male OBTAINS

Line 100 - it by patrolling its ranges – you mean by patrolling along it (the territory)? – the English really need to be revised by the native speaker

Line 102 - either patrol larger sections – patrol OVER the larger sections

Line 104 - when a male focus on a selected area but do not own a territory and hence do not defend it à when a male focusES on a selected area but DOES not own a territory and hence DOES not defend it

Line 105 - Once a female approach à Once a female approachES

Figure 1 – do you maybe have some more quality photo of the species, e.g. something more up to close?

Line 112 - Behavioural data were collected in 2016 – when in 2016? Please specify (e.g. in summer 2016)

Line 113 – it would be good to show your localities on the map

Line 137 - were measured à WAS measured

Table 1 – please revise the coordinates (e.g. number of decimal places, species between the numbers)

Line 158 - in order to facilitated – in order to facilitate

Lines 159-160 - Statistical significance was established at P > 0.05 – shouldn’t this be P < 0.05

Line 163 - body size were positively correlated with latitude – WAS not were – okay, I am gonna stop correcting the language, please let the native speaker revise the whole MS

Table 2 – please put the species name in italic. Also, in tables and figures it is good to use the full species name, not the abbreviation

Line 203 - Whereas, courtship is shaped by other factors more important than breeding season time constraints – such as, and why?

Lines 206, 207, 252 – please add authority

Line 217 – what is the threshold identified here, please specify while discussing

Line 222 - Based on these studies, our results suggest that high latitude populations of C. splendens have high capacity to expand north because of their bold behaviour – aren’t those populations already at the north side of the distribution range?

Lines 267-268 - Body size differences along the latitudinal gradient could be explained by variable voltinism in the study species… I would suggest adding the information about the differences in voltinism, e.g. Differences in body size along the latitudinal gradient could be explained by variable voltinism in the studied species (i.e. semivoltine in boreal and univoltine in other regions of its distributional range) (Dijkstra et al. 2022; Schmidt-Kloiber & Hering 2015). This depends on thermal conditions, growth season length and seasonal regulation of nymphal development [52,53].

  • Dijkstra, K.-D.B., Wildermuth, H. & Martens, A. (2022): Dataset "Odonata". www.freshwaterecology.info - the taxa and autecology database for freshwater organisms, version 8.0 (accessed on 09.02.2022).
  • Schmidt-Kloiber, A. & Hering D. (2015): www.freshwaterecology.info - an online tool that unifies, standardises and codifies more than 20,000 European freshwater organisms and their ecological preferences. Ecological Indicators. https://doi.org/10.1016/j.ecolind.2015.02.007

Line 269, 272 – I would suggest using the term nymph instead of larva as the latter is generally used for holometabolous insects, which Odonata are not

Author Response

Dear Reviewer,

We appreciate the time and effort that you have dedicated to your feedback on
our manuscript titled "Cross-latitude behavioural axis in an adult damselfly Calopteryx splendens (Harris, 1780)". 

Below are listed (in red) our responses to all the comments. 

The study of Golab et al. entitled Cross-latitude behavioural axis in an adult damselfly Calopteryx splendens investigated variation and mutual relationship of three behavioural traits (activity, courtship and boldness) of the damselfly species, C. splendens, along a latitudinal gradient (between Sweden, Poland and Hungary). The authors recorded a clear latitudinal gradient in species activity, while two other behaviour traits, courtship and boldness, might be more significantly influences by the local factors. Studies investigating multiple behavioural traits and behavioural syndromes in natural populations of temperate insects along a latitudinal gradient are rare and those results are interesting and novel, and I would love to see them published. Nevertheless, there are several points that need to be addressed, and one of them is the English language revision by the native speaker. In some cases, I was not completely sure what the authors wanted to say, I corrected some parts/suggested some modifications, but there are still many left to be revised. Additionally, some comments and suggestions are presented below:

Title – please put the species name in italic, and add authority

  • corrected

Line 16 – from southern to northern region

  • corrected, line 18

Line 17 - This because northern animals – This IS because...

  • corrected, line 19

Line 18 - Activity was the only behaviour positively associated.... – I would rephrase this as Activity was the only behaviour FEATURE positively associated...

  • corrected, line 21

Lines 19-20 - Courtship was highest in the central region whereas boldness was highest in the north but did not differ between central and south. – what does this mean? What does the highest courtship, of the highest boldness mean?

  • corrected, line 22-23

Lines 20-21 - An activity–boldness syndrome was only present in the southern and northern region, and courtship was associated with boldness in low latitudes only. – it would be good to refer to your regions in one way within the sentence, i.e. if you speak about the southern populations when explaining one behavioural trait, then do not mention low latitude when you are talking about some other investigated trait.

  • Corrected, lines 23-26

Lines 23 - which not always follow latitudinal gradient. – you mean which do not always follow latitudinal gradient?

  • corrected, line 28

Line 26 - Latitudinal differences in behaviour has received relatively limited interest – Latitudinal differences in behaviour HAVE received—also, you mean research interest?

  • corrected, line 32

Line 30 – I would suggest using the authority when you mention the species for the first time, in abstract and in the text

  • corrected, line 36-37

Keywords: I would suggest replacing the species name, Calopteryx splendens, with Odonata, as you already have species name in the title

  • Thank you for this suggestion, replaced, line 51

Line 50 - Cross-population studies of behavioural traits have gained large interest – better say have gained an extensive research interest….

  • corrected, line 67-68

Line 50 - These within species studies – what does this mean, please rephrase

  • done, line 68

Line 52 - Behavioural comparisons of specific interest are those along different environmental gradients – you mean Specifically interesting are behavioural comparisons along different environmental gradients?

  • rephrased, line 70

Line 66 - invasive signal crayfish – please add latin species name in the brackets

  • The example of the crayfish was removed from the introduction and replaced by an example of an insect species, line 84-87. We think that updated citation is a better fits to the target journal.

However, absence of behavioural syndromes (e.g. activity – boldness) have also been reported [19]. In that study the traits were decoupled by different strengths of natural and sexual selections in different environments [20]. – when you say „in that study“ you refer to the reference no. 19 or 20?  

  • The last sentence should be referred to the both references (no. 9 and 25 now). Because reference no. 25 additionally explains the mechanisms of the decoupling of behavioural syndromes, line 89-91

Lines 64 – 80 – are there maybe similar examples in aquatic insects?

  • There are not many invasive species in aquatic insects (Fenoglio et al. 2016). However, we replaced the crayfish with an example on vespid species, lines 84-87

Line 84 - and conspicuous behavioural displays… - you mean and displays conspicuous behaviour?

  • corrected, line 103

Lines 93-94 – please add reference

  • done, line 110

Line 95 - the species experience à the species experiences, or goes through

  • corrected, line 116

Line 96 - The species exhibit strong site fidelity – you are talking about one species, so it has to be the species EXHIBITS

  • corrected, line 117

Line 100 – a male OBTAINS

  • corrected, line 120

Line 100 - it by patrolling its ranges – you mean by patrolling along it (the territory)? – the English really need to be revised by the native speaker

  • corrected, line 120

Line 102 - either patrol larger sections – patrol OVER the larger sections

  • corrected, line 123

Line 104 - when a male focus on a selected area but do not own a territory and hence do not defend it à when a male focusES on a selected area but DOES not own a territory and hence DOES not defend it

  • corrected, line 124

Line 105 - Once a female approach à Once a female approachES

  • corrected, line 126

Figure 1 – do you maybe have some more quality photo of the species, e.g. something more up to close?

  • Figure 1 changed

Line 112 - Behavioural data were collected in 2016 – when in 2016? Please specify (e.g. in summer 2016)

  • Added, line 135

Line 113 – it would be good to show your localities on the map

  • Figure 1 contains the map now

Line 137 - were measured à WAS measured

  • corrected, line 171

Table 1 – please revise the coordinates (e.g. number of decimal places, species between the numbers)

  • done, line 175

Line 158 - in order to facilitated – in order to facilitate

  • corrected, line 195

Lines 159-160 - Statistical significance was established at P > 0.05 – shouldn’t this be P < 0.05

  • corrected, line 197

Line 163 - body size were positively correlated with latitude – WAS not were

  • corrected, line 211

Table 2 – please put the species name in italic. Also, in tables and figures it is good to use the full species name, not the abbreviation

  • corrected, Table 1, 2

Line 203 - Whereas, courtship is shaped by other factors more important than breeding season time constraints – such as, and why?

  • Sentence rephrased, lines 250-252. The discussion on “why” can be found in the text below, lines 306-318

Lines 206, 207, 252 – please add authority

  • Done, line 254, 256, 272

Line 217 – what is the threshold identified here, please specify while discussing

  • Done, line 266

Line 222 - Based on these studies, our results suggest that high latitude populations of C. splendens have high capacity to expand north because of their bold behaviour – aren’t those populations already at the north side of the distribution range?

  • Yes, we sampled at the northern distribution range, however, recent studies suggest that the species expands northwards (Hickling and Roy 2005) and we assume the expansion may continue.

Lines 267-268 - Body size differences along the latitudinal gradient could be explained by variable voltinism in the study species… I would suggest adding the information about the differences in voltinism, e.g. Differences in body size along the latitudinal gradient could be explained by variable voltinism in the studied species (i.e. semivoltine in boreal and univoltine in other regions of its distributional range) (Dijkstra et al. 2022; Schmidt-Kloiber & Hering 2015). This depends on thermal conditions, growth season length and seasonal regulation of nymphal development [52,53].

  • rephrased, lines 319-324

Line 269, 272 – I would suggest using the term nymph instead of larva as the latter is generally used for holometabolous insects, which Odonata are not.

  • In odonate literature (e.g. Corbet 2004; Córdoba-Aguilar 2008) it is customary to use term “larva”, and not “nymph”. We decided to keep term „larva”.

- References

Corbet P. 2004. Dragonflies: behaviour and ecology of Odonata. Harley Books.

Córdoba-Aguilar A. 2008. Dragonflies and Damselflies: Model Organisms for Ecological and Evolutionary Research. Oxford University Press, USA.

Fenoglio S, Bonada N, Guareschi S, López-Rodríguez MJ, Millán A, Tierno de Figueroa JM. 2016. Freshwater ecosystems and aquatic insects: a paradox in biological invasions. Biology Letters. 12(4):20151075. doi:10.1098/rsbl.2015.1075.

Hickling R, Roy DB. 2005. A northward shift of range margins in British Odonata. Glob Chang Biol. 11:502–506.

Reviewer 2 Report

The manuscript presented here focuses on an interesting topic – do behavioural traits have any relationships with latitudinal gradients, vis-à-vis climatic gradients. The results presented here reaffirms our ideas of odonatan activity; odonates are active for longer in hotter regions. This is not to say that boldness and mating activity follow a similar trend. The article is commendable, although some improvements can be made to make the story a bit stronger.

Abstract

I think it would be very useful if the authors can include their study area in the abstract. At the moment, it is unclear exactly where the study was done.

Introduction

An interesting question that comes to mind is whether climate drives behaviour, or does certain behaviours drive responses to certain climatic conditions? Many authors argue that behavioural responses are a product of climate, and not the other way round. Can the authors perhaps explore this a bit in their introduction?

Although these are good examples of trait responses, do the authors perhaps have some examples specific to odonatology? I think by including these, a much more inclusive background image will be sketched.

Methods

How many river sections were included in the study? Was it multiple per geographic area, or only one per geographic area? This needs clarification.

Results

I am wondering what the results would have looked like if the authors used the actual latitudinal values instead of north vs. central vs. south. Were the correlations perhaps stronger or weaker? (This would only be possible if there were multiple river stretches per geographic area).

The axes of figure 1 need to be checked, and maybe enlarged a bit. These figures are quite difficult to read.

Perhaps the section on behavioural syndromes should be given at the start of the results? Then the readers know straight away how the traits were related.

Discussion

I can’t help but wonder whether odonate taxonomy plays a great role here – wouldn’t anisopterans be inherently i.e. more active/bold compared to zygopterans? I know you control for this by using only one species, but what do you think of other odonate families? Would they show the same trends?

While I think your overall research question was strong, and importantly, simple enough to test, I think that one major limitation of your study was that your geographic areas was fairly narrow. Do the authors think that patterns may have been stronger over a wider geographic area?

Author Response

Dear Reviewer,

Thank you for your time spent reviewing the manuscript, and your opinions regarding the science and presentation of the material. Our responses are listed below (in red).

The manuscript presented here focuses on an interesting topic – do behavioural traits have any relationships with latitudinal gradients, vis-à-vis climatic gradients. The results presented here reaffirms our ideas of odonatan activity; odonates are active for longer in hotter regions. This is not to say that boldness and mating activity follow a similar trend. The article is commendable, although some improvements can be made to make the story a bit stronger.

 Abstract 

I think it would be very useful if the authors can include their study area in the abstract. At the moment, it is unclear exactly where the study was done.

  • Done, line 35-36

 Introduction

 An interesting question that comes to mind is whether climate drives behaviour, or does certain behaviours drive responses to certain climatic conditions? Many authors argue that behavioural responses are a product of climate, and not the other way round. Can the authors perhaps explore this a bit in their introduction?

  • We also lean toward the argument that environment (climate as one of environmental factors) shapes behavioural traits. We expanded the first paragraph of the introduction, lines 60-66

Although these are good examples of trait responses, do the authors perhaps have some examples specific to odonatology? I think by including these, a much more inclusive background image will be sketched.

  • We are not sure what part of the Introduction this question refers to. We need more specific question/instructions here in order to improve the introduction in this regard. However, in general experimental studies on behavioural responses of odonate species to environmental factors are scarce.

Methods

 How many river sections were included in the study? Was it multiple per geographic area, or only one per geographic area? This needs clarification.

  • Clarified, line 139

Results

 I am wondering what the results would have looked like if the authors used the actual latitudinal values instead of north vs. central vs. south. Were the correlations perhaps stronger or weaker? (This would only be possible if there were multiple river stretches per geographic area).

  • We sampled one river section per population, and two populations per region. However, the two populations per region were separated by several kilometres. Hence, the latitudinal values are almost the same in each geographic region (see Tab. 1 and Fig. 1b). 

The axes of figure 1 need to be checked, and maybe enlarged a bit. These figures are quite difficult to read.

  • Corrected

Perhaps the section on behavioural syndromes should be given at the start of the results? Then the readers know straight away how the traits were related.

  • Section moved, lines 204-209

Discussion

 I can’t help but wonder whether odonate taxonomy plays a great role here – wouldn’t anisopterans be inherently i.e. more active/bold compared to zygopterans? I know you control for this by using only one species, but what do you think of other odonate families? Would they show the same trends?

  • We think that some species would show different patterns here, especially if one takes into account the absolute values. Odonates adopt variety of life strategies, with typical perchers and species spending most of their time in active flight (i.e., patrolling). With regard to boldness, again the variety of life styles and environments inhabited by different species allow us to assume that we would also find bolder species among anizopterans than in zygopterans. For example, comparing Orthetrum coerulescens, a dragonfly species that prefers perching over active flight with calopterygids, which are quite mobile and devote much of their time for aerial contests, would bring contrasting results.

 While I think your overall research question was strong, and importantly, simple enough to test, I think that one major limitation of your study was that your geographic areas was fairly narrow. Do the authors think that patterns may have been stronger over a wider geographic area?

  • We agree that data from southernmost populations would bring more complete picture of the geographic pattern in the studied behaviours. We think that populations from the southern range margin would differ in the behavioural profiles from the other latitudes. One of the reasons for this could be that the species has an ability to change its voltinism (additional generation within a growth season). The more generations within a season, the stronger seasonal time constraints for larval growth and development. Faster growth and development commonly goes in pair with higher activity rate, boldness, etc. in larval as well as adult stage. It has been reported that traits in larval and adult stage are not decoupled through metamorphosis (Brodin 2009). On the other hand, southernmost populations probably undergo climatic constraints, for example very high air temperatures during breeding season. This in turn, can reduce adult activity during the hottest period of the day and act as a serious time constraint. Our previous study on Lestes sponsa sexual behaviours across broad latitudinal gradient (covering northern and southern range margins and central populations, Golab et al. 2019) showed that southern populations significantly differed from central and northern ones. However, sponsa is an obligatory univoltine species, hence climatic constraints are more severe for the species, than would be for C. splendens, which has a variable voltinism.

  • References 

Brodin T. 2009. Behavioral syndrome over the boundaries of life carryovers from larvae to adult damselfly. Behav Ecol. 20(1):30–37. doi:10.1093/beheco/arn111.

Golab M, Johansson F, Sniegula S. 2019. Let’s mate here and now – seasonal constraints increase mating efficiency. Ecological Entomology. doi:10.1111/een.12739.

Reviewer 3 Report

Overall, I thought this was an interesting paper. Latitude differences in behavior are certainly an intriguing topic, and I, like the authors, I have not seen this investigated before. I do think this is a potentially publishable paper, but there are a few issues that need to be addressed first. I believe these issues could be addressed in a revision or resubmission. I do not believe additional data needs to be collected.

Major concerns

1) The simulated predatory attack needs further clarification. Where did the artificial bird appear from? How did it move toward the subject? How fast did it move? How far away did it start? How close did it come to the subject? How long was the duration of the attack? Did the artificial bird move toward the subject and immediately retract? Did it move toward the subject and remain for the observation period? These are all important details. Without understanding the details of the predator attack, it also becomes hard to understand the immobility, first move, and time to return measurements. For example, my current understanding is that immobility and first move will always be the same value, as immobility is the duration of no movement at the start of an observation, and first move is the latency of movement at the start of an observation.

2) I have several concerns related to the analysis. Many parts of the analysis can be seen as arbitrary. The authors need to provide statistical and/or theoretical reasons why their raw data could only be reasonably analyzed and interpreted in the manner that they do. I want to clarify that I don’t suspect there are any issues with the data or that the authors are being misleading. My concern is that the analysis could potentially be performed another way to obtain very different results, and there is not adequate information to support the current analysis.

Why were the measures for activity and body size averaged to create a final measurement for the GLMM analysis while the boldness score was derived from PCA?

Why would the authors assume that behaviors during a simulated predatory attack are a measure of boldness? Other potential causes could be differences in visual ability, momentary differences in energy or body temperature. It does not seem parsimonious to assume a cause for these behaviors without additional information.

If a related cause is not known for the boldness behaviors, why cluster the behaviors together assuming they are all related? Some additional theory, literature or analysis is needed to support these categorizations.

It does appear that the boldness PCA suggests the boldness behaviors are correlated, but as activity and courtship measures were not simultaneously considered in the PCA, is there any reason to suspect that a measurement like number of patrol flights is not related to first move on a predatory attack? Could these behaviors be grouped differently? If all measures were included in a PCA, then I believe, we could at least know if there was a statistical reason to group measurements together, even if there was no a priori theoretical reason to do so.

If we follow this line of consideration, it may be that classification of behavior as activity vs boldness is completely arbitrary. In their analysis, the authors found that activity and boldness were correlated in southern and northern regions, and that courtship was correlated with boldness in the southern region. How would these results change if the categories were redefined? How do we know if the current categorization of behaviors is correct?

I believe the answer to this is to include all measures in the PCA, and use the results to determine functional categories, then proceed to the GLMM. I am open to considering other solutions as well. I would be especially pleased if the authors can support their categorization of measurements both statistically and theoretically.

Minor comments

1) The first few paragraphs of the introduction and the first few references are a little ambiguous. The statements are not well-defined, and since many citations are entire books, it is hard to know what the authors mean, or if the citations actually support their intent. For example, the second line suggests that behavior evolves faster than other traits. I first took that as a misstatement and thought that perhaps the authors meant that behavior changes through learning and other ontogenetic methods faster than evolution can occur. Then, I considered that perhaps the authors meant exactly as they wrote, and the citations would provide clarification. They did not. This is just one example. Later paragraphs in the introduction do not seem to have this issue. Toward the end of the introduction, many specific examples are given, and the citations seem appropriate. Please address the minor lack of clarity and use of references in the early parts of the introduction

2) Line 86 – The authors discuss their hypothesis with C. splendens and suggest that behavior will be correlated with latitude. This is clear, but the statement “with progressively higher values northward” is not clear. Higher values of what? No values for behavior have been defined yet.

3) The authors define behavioral syndrome as a correlated cluster of behavioral traits. However, the difference between behavior, behavioral trait, behavioral expression, and behavioral axis are not clarified. The authors may intend some of these words to be synonyms, but that is not always true for all parts of the animal behavior and personality fields, so it is important to clarify.

4) line 100 – obtains

5) line 105 – approaches

6) The focal species is described primarily in the methods, which seems to be introduction material.

7) Reporting basic descriptive statistics for all measures before describing the results of the GLMM from derived measures would be useful.

8) Table 2. Post-hoc comparisons. The label behavioral axis is confusing as body size is not behavior.

9) Multiple tables are labeled as Table 2

Author Response

Dear Reviewer,

We greatly appreciate the thorough and thoughtful comments provided on our submitted article entitled Cross-latitude behavioural axis in an adult damselfly Calopteryx splendens (Harris, 1780). We replied in detail to all the comments (in red). Please see the attachment.

If you still have any questions or concerns about the manuscript, we will be happy to address them, now in a timely manner.

Round 2

Reviewer 3 Report

The authors suitably responded to all my concerns.

Author Response

Dear Reviewer,

Thank you for accepting our replies to your comments.